# Semifinite von Neumann algebras in gauge theory and gravity

Shadi Ali Ahmad,[1] Marc S. Klinger,[2] and Simon Lin[1]

[1] *Center for Cosmology and Particle Physics, New York University, New York, NY 10003, USA*
[2] *Illinois Center for Advanced Studies of the Universe & Department of Physics,*
*University of Illinois, 1110 West Green St., Urbana IL 61801, U.S.A.*

(Dated: July 3, 2024)

von Neumann algebras have been playing an increasingly important role in the context of gauge theories and gravity. The crossed product presents a natural method for implementing constraints through the commutation theorem, rendering it a useful tool for constructing gauge invariant algebras. The crossed product of a Type III algebra with its modular automorphism group is semifinite, which means that the crossed product regulates divergences in local quantum field theories. In this letter, we find a sufficient condition for the semifiniteness of the crossed product of a type III algebra with *any* locally compact group containing the modular automorphism group. Our condition surprisingly implies the centrality of the modular flow in the symmetry group, and we provide evidence for the necessity of this condition. Under these conditions, we construct an associated trace which computes physical expectation values. We comment on the importance of this result and and its implications for subregion physics in gauge theory and gravity.

## I. INTRODUCTION

In quantum field theory, one can assign an algebra of observables to any open set of spacetime which may be weak completed in a Hilbert space to a von Neumann algebra [1–3]. Since von Neuman algebras admit a classification into factors of different types organized by the properties of their lattice of projections [4–9], a natural question to ask is: what is the type of algebras of observables appearing in quantum field theory? The answer was given in seminal works [10–16], where it was argued that generically such algebras are Type III$_1$. One striking property of such algebras is that they are not semifinite, which introduces a plethora of divergences in the computation of physical observables and entropies [17–23]. As we expect such quantities should be finite, this calls for the construction of a semifinite trace affiliated with the algebra, whose existence is tied to the regulation of these divergences.

The structure theory of Type III$_1$ factors was analyzed early on by Takesaki [24], where it was shown that the crossed product of such an algebra with its modular automorphism group is semifinite, even though the original algebra is not. The modular crossed product then allows us to access the semifinite features of the original algebra. Indeed, this result applied to physical scenarios has recently given us an understanding of generalized entropy from a Lorentzian and canonical picture [25–31].

By the commutation theorem and its generalizations [24, 32–39], the crossed product algebra respects natural constraints associated with the group used to form it. Recent progress in understanding the relevance of the crossed product algebra in the context of gauge theory and gravity has suggested that, in some cases, one can interpret it as the gauge-invariant algebra of observables of the theory [30, 40]. Thus, one has incentive to perform the crossed product of algebras of observables by groups that are not necessarily the modular automor-

phism group [29, 30, 40, 41].[1] However, the semifiniteness of crossed products with more general groups has, to our knowledge, not been considered in previous literature with the exception of Ref. [43] which considers direct products between the modular automorphism group and a compact group. Without a semifinite trace on the gauge-invariant algebra of observables, one runs again into familiar divergences from quantum field theory when computing entropic observables.

In this letter, we provide a sufficient condition guaranteeing the semifiniteness of the crossed product of a Type III algebra with a locally compact group. Our result puts the crossed product on more solid physical ground as it allows for the definition of a semifinite trace in gauge theory and gravity. Our Theorem 2 leverages the theory of dual weights [46–49] which allows one to understand the modular structure of the crossed product in terms of that of the original algebra [46–49]. We phrase our result in terms of physical conditions, namely the existence of a suitably invariant weight on the original algebra satisfying the KMS condition relative to a subgroup of the relevant symmetry [50–52]. We then construct a trace on the resultant semifinite algebra which reduces to the known expression in the modular case. Our condition has a surprising implication: the modular flow must be a central subgroup of the symmetry group. We investigate this further and prove Theorem 3 showcasing the *necessity* of this centrality in some cases.

## II. CROSSED PRODUCTS WITH LOCALLY COMPACT GROUPS

In this section, we review crossed products with locally compact groups, following Ref. [46]. The building block

---

[1] Different perspectives on the crossed product in the context of quantum reference frames appear in Refs. [42–45].

is the notion of a dynamical system.

**Definition 1.** *A dynamical system is a triple $(M, G, \alpha)$ where $M$ is a von Neumann algebra[2], $G$ is a locally compact group, and $\alpha : G \to \mathrm{Aut}(M)$ is an action of the group on the algebra via automorphisms.*

Examples of such an object are algebras of observables of quantum field theories localized to subregions of spacetime [1] where $G \simeq \mathbb{R}_{\mathrm{mod}}$ is the modular automorphism group and $\alpha = \sigma^\phi$ is the modular action associated to some weight $\phi : M \to \mathbb{C}$.

To work in a standard quantum picture, we need to represent a dynamical system on a Hilbert space. For any dynamical system $(M, G, \alpha)$ with a given Hilbert space representation $\pi : M \to B(H)$, there exists a canonical extension of $(\pi, H)$ representing both the algebra $M$ and the group $G$ on an extended Hilbert space $L^2(G, H) \simeq H \otimes L^2(G)$ where the automorphism $\alpha$ is implemented unitarily.

**Definition 2.** *The canonical covariant representation of a dynamical system $(M, G, \alpha)$ induced from the representation $\pi : M \to B(H)$ is a triple $(L^2(G, H), \pi_\alpha, \lambda)$ where $\pi_\alpha : M \to B(L^2(G, H)))$ is a representation of $M$ on $L^2(G, H)$, and $\lambda : G \to U(L^2(G, H)))$ is a unitary representation of $G$ on the space given by*

$$\pi_\alpha(x)\xi(g) := \pi \circ \alpha_{g^{-1}}(x)\xi(g), \quad \lambda(h)\xi(g) := \xi(h^{-1}g)$$

*for any $\xi \in L^2(G, H)$, $g \in G$, and $x \in M$. The representation is covariant in the sense that the automorphism $\alpha$ is unitarily implemented: $\pi_\alpha \circ \alpha_g = Ad_{\lambda(g)} \circ \pi_\alpha$.*

The two representations combine into a new von Neumann algebra called the crossed product.

**Definition 3** (Crossed product). *Let $(L^2(G, H), \pi_\alpha, \lambda)$ be the canonical covariant representation of the dynamical system $(M, G, \alpha)$ induced from the representation $\pi : M \to B(H)$. The crossed product of $M$ with $G$ relative to the automorphism $\alpha$ is the von Neumann algebra*

$$M \rtimes_\alpha G := \{\pi_\alpha(M), \lambda(G)\}'', \tag{1}$$

*where the double-prime denotes weak closure in $L^2(G, H)$.*

One useful way of understanding the crossed product is through the generalization of Takesaki duality which we refer to as the commutation theorem [24, 32–39].

**Theorem 1** (Commutation theorem). *Let $\tilde{M} = M \otimes B(L^2(G))$ where $M$ is part of a dynamical system $(M, G, \alpha)$ and define by $\tilde{\alpha} := \alpha \otimes Ad\lambda_r$ an extension of the automorphism $\alpha$ on $M$ to $\tilde{M}$ with $\lambda_r$, the right regular representation of $G$. Then, the fixed-point subalgebra of $\tilde{M}$ under $\tilde{\alpha}$ is isomorphic to $M \rtimes_\alpha G$.*

The commutation theorem allows us to understand the crossed product as an algebra respecting particular constraints imposed by the group $G$. This construction can be applied to general gauge theories [30, 40] in which case the crossed product is the gauge-invariant algebra of observables. In [40] this result has been directly related to more conventional approaches to constraint quantization such as refined algebraic quantization (RAQ) [54–58], the BRST method [59–67], and the Faddeev-Popov approach in path integral quantization [68–70]. Briefly, the group generators included into the algebra via Eq. (1) furnish apparently gauge non-invariant degrees of freedom which are nonetheless necessary to achieve gauge invariance in the full theory. In RAQ, these degrees of freedom facilitate the group averaging of operators, in BRST they equip the algebra with a representation of the appropriate group cohomology, and in the Faddeev-Popov approach they are used to construct an insertion which localizes the path integral to the constraint surface. For different perspectives on the crossed product in the context of quantum references frames and constraint quantization, see [44, 45, 71].

## III. SEMIFINITENESS OF CROSSED PRODUCTS

Here, we review dual weights as necessary to prove our main result. In the following $(M, G, \alpha)$ is a dynamical system and $(L^2(G, H), \pi_\alpha, \lambda)$ is the canonical covariant representation induced from the representation $\pi : M \to B(H)$. We let $\omega \in M_*$ be a faithful, semifinite, normal weight on the original algebra $M$, and denote by $\omega_g \equiv \omega \circ \alpha_g \in M_*$ the family of 'G-translates' of $\omega$. We denote by $\Delta_{g,h} : H \to H$ the relative modular operator and by $u_t^{g,h} \equiv \Delta_{g,h}^{it} \Delta_{h,h}^{-it} \in M$ the Connes cocycle of the weights $\omega_g$ and $\omega_h$. The modular operator of the untranslated weight $\omega$ is given by $\Delta \equiv \Delta_{e,e}$. Dual weights are a natural way to associate the weight $\omega$ with a faithful, semifinite, normal weight $\tilde{\omega} \in (M \rtimes_\alpha G)_*$ whose modular data is encoded entirely in the modular data of $\omega$ and its $G$-translates.

**Definition 4** (Haagerup's algebra). *Given a dynamical system $(M, G, \alpha)$ the algebra $M_G$ is a generalization of the group algebra which consists of continuous compactly supported maps from $G$ to $M$. $M_G$ is a $*$-algebra equipped with a product $\star$*

$$(\mathfrak{X} \star \mathfrak{Y})(g) \equiv \int_G \mu(h) \, \alpha_h(\mathfrak{X}(gh))\mathfrak{Y}(h^{-1}),$$

*where $\mu(h)$ is the left-invariant Haar measure on $G$, and involution $\sharp$*

$$\mathfrak{X}^\sharp(g) \equiv \delta(g^{-1})\alpha_{g^{-1}}(\mathfrak{X}(g^{-1}))^*.$$

The algebra $M_G$ can be interpreted as encoding the orbit data of operators in $M$ under the action $\alpha$. It

---

[2] See [53] for a recent physical review of some elements of von Neumann algebras.

can also be regarded as providing an alternative construction of the crossed product. The covariant representation $(L^2(G,H), \pi_\alpha, \lambda)$ induces a *-representation $\rho : M_G \to B(L^2(G,H))$ given by

$$\rho(\mathfrak{X}) \equiv \int_G \mu(g)\ \lambda(g)\pi_\alpha(\mathfrak{X}(g)). \tag{2}$$

The algebra $\rho(M_G) \subset B(L^2(G,H))$ is weakly dense in the crossed product $M \rtimes_\alpha G$ and thus $\rho(M_G)'' = M \rtimes_\alpha G$.

Having defined the algebra $M_G$[3], we can now state the definition of the dual weight:

**Definition 5** (Dual weight). *Given a faithful, semifinite, normal weight $\phi \in M_*$ there exists a unique faithful, semifinite, normal weight $\tilde{\phi} \in (M \rtimes_\alpha G)_*$ satisfying:*

- *For all[4] $\mathfrak{X} \in M_G$,*

$$\tilde{\phi}\left(\rho(\mathfrak{X}^\sharp \star \mathfrak{X})\right) = \phi\left((\mathfrak{X}^\sharp \star \mathfrak{X})(e)\right). \tag{3}$$

- *For all $\xi \in L^2(G,H)$ the dual modular operator, namely the modular operator of the weight $\tilde{\phi}$, $\tilde{\Delta} : L^2(G,H) \to L^2(G,H)$ is given by*

$$\left(\tilde{\Delta}^{it}\xi\right)(g) = \delta(g)^{it}\pi(u_t^{g,e})\Delta^{it}\left(\xi(g)\right). \tag{4}$$

The construction of the dual weight is due originally to Digernes [47] for the separable case and was later refined by Haagerup [48, 49] for the general case.

Before stating our results, we make one final definition.

**Definition 6** (Quasi-invariance). *Let $(M,G,\alpha)$ be a dynamical system and $\omega$ be a faithful, semifinite, normal weight on $M$. If $\omega \circ \alpha_g = \delta(g)^{-1}\omega$, we call $\omega$ quasi $\alpha$-invariant.*

While the condition $\omega \circ \alpha_g = \delta(g)^{-1}\omega$ may seem physically unnatural at first sight, we see that in the typical case of a unimodular group, this reduces to requiring the $\alpha$-invariance of $\omega$. Moreover, it could be interpreted as the suitable notion of $\alpha$-invariance for non-unimodular groups [56, 72–74]. The dual modular operator of a quasi $\alpha$-invariant weight $\omega$ can be simplified by observing $u_t^{g,e} = \delta(g)^{-it}\mathbb{1}$, which implies $\tilde{\Delta} = \Delta \otimes \mathbb{1}$. The form of this dual modular operator motivates the following theorem:

**Theorem 2** (Semifiniteness). *Let $(M,G,\alpha)$ be a dynamical system with $(\mathbb{R},+) < G$ a subgroup. We denote by $\gamma : \mathbb{R} \hookrightarrow G$ the inclusion of this subgroup. If (i) there exists a faithful, semifinite, normal weight, $\omega \in M_*$ that is quasi $\alpha$-invariant and (ii) the weight $\omega$ is KMS with respect to the action of $\mathbb{R} < G$, then $M \rtimes_\alpha G$ is semifinite.*

---

[3] We refer the reader to Appendix A for more detail.
[4] To be completely rigorous, this should be stated in terms of the product of $M_G$ with the ideal $\mathfrak{n}_\phi \equiv \{x \in M \mid \phi(x^*x) < \infty\}$.

*Proof.* By the uniqueness of the KMS condition, assumption (ii) implies that the modular automorphism $\sigma_t = \alpha_{\gamma(t)}$.

Assumption (i) implies that the dual modular automorphism $\tilde{\sigma}$ is implemented by $\sigma \otimes \mathbb{1}$. It moreover implies by $\sigma^{\omega_g} = \alpha_g^{-1} \circ \sigma^\omega \circ \alpha_g$ that the modular flow $\sigma$ commutes with the automorphism $\alpha$ [75]. In this case, $\sigma \otimes \mathbb{1}$ is inner implemented in the crossed product $M \rtimes_\alpha G$ as proved in Lemma 2 of Appendix D. Then, $\tilde{\sigma}$ is also inner implemented.

A von Neumann algebra is semifinite if and only if it admits a semifinite normal weight for which its modular automorphism group is innerly implemented (c.f. Theorem 3.14 in Ref. [46]), which proves our result. □

A few remarks are in order:

1. An advantage of quasi $\alpha$-invariance is that one does not need to assume that the group von Neumann algebra $\mathcal{L}(G)$ is semifinite to obtain the semifiniteness of $M \rtimes_\alpha G$. If one had instead required $G$-invariance of $\omega$, the dual modular automorphism would have included a factor of the module function, which then requires innerness of the adjoint action of the multiplication operator induced by the module function. This is implied by the semifiniteness of $\mathcal{L}(G)$ in Appendix B, where it is shown that this operator is precisely the modular operator for the Plancherel weight of the group von Neumann algebra.

2. The assumptions of our Theorem 2 imply that $\mathbb{R} < G$ is a central subgroup. The centrality of $\mathbb{R}$ seems to be a necessary condition in some cases as we record in the following theorem:

**Theorem 3** (Centrality). *Let $(M,G,\alpha)$ be a dynamical system with $M$ a type III algebra. Suppose that $M$ admits a faithful, semifinite, normal weight $\omega$ which is KMS with respect to the action of a subgroup $\mathbb{R} < G$. Moreover, assume that the relative commutant $\pi_\alpha(M)^c \equiv \pi_\alpha(M)' \cap (M \rtimes_\alpha G)$ is trivial. Then, the crossed product $M \rtimes_\alpha G$ is semifinite if and only if $\omega$ is quasi $\alpha$-invariant and so $\gamma(\mathbb{R})$ is a central subgroup.*

*Proof.* Suppose $M \rtimes_\alpha G$ is semifinite, which means that the modular automorphism of any faithful, semifinite, and normal weight is inner implemented [46]. In particular, this must hold for $\tilde{\omega}$, the dual weight of $\omega$. In Appendix E, we show that under the assumption that $\pi_\alpha(M)^c = \mathbb{C}$, $\mathrm{Ad}_{\lambda(\gamma(t))}$ is the unique inner implementer of the dual modular automorphism $\tilde{\sigma}$ in $M \rtimes_\alpha G$. Now, compare the action of $\mathrm{Ad}_{\lambda(\gamma(t))}$ and $\tilde{\sigma}_t$ on an element $\lambda(g)$. The former simply gives rise to $\lambda(\gamma(t)g\gamma(t)^{-1})$, while the latter is given by Eq. (D2). Forcing them to be equal implies that

$$\lambda(g^{-1}\gamma(t)g\gamma(t)^{-1}) = \delta(g)^{it}\pi_\alpha[u_t^{g|e}].$$

The operators on either side of the above equation are *distinct* generators of the crossed product and so cannot be equal unless they are proportional to the identity. The only way for the operator on the left to be trivial is if $\gamma(\mathbb{R})$ is central in $G$. Finally, the triviality of the operator on the right implies that $\omega \circ \alpha_g = \delta(g)^{-1}\omega$ since the cocycle is proportional to the identity if and only if the two weights are proportional. In other words, the weight $\omega$ is quasi $\alpha$-invariant.

The reverse direction is proved by our Theorem 2. □

3. Once it has been established that the crossed product is semifinite under the assumptions of Theorem 2, it is manifest that the modular flow of $\tilde{\psi} \in (M \rtimes_\alpha G)_*$ is inner implemented for every $\psi \in M_*$. If the weight $\psi$ is not quasi $\alpha$-invariant, its modular flow may be implemented by a non-central subgroup of $G$ since the action of its modular group is related to that of $\omega$ by conjugation with cocycles. This conjugation can twist the embedding of $\mathbb{R}$ in $G$, rendering the subgroup non-central. This should be contrasted with Theorem 3 which further assumes the triviality of $\pi_\alpha(M)^c$ and therefore forces the modular flow of any KMS weight relative to $\alpha|_\mathbb{R}$ to be central.

## IV. CONSTRUCTING TRACES

A semifinite von Neumann algebra admits a tracial weight. In Theorem 2 and Appendix D we show that the action of the dual modular flow is innerly implemented by a unitary operator $A^{it}$. In particular,

$$\tilde{\sigma}_t = \mathrm{Ad}_{A^{it}}, \quad A^{it} = \mathbb{1} \otimes \lambda(\gamma(t)), \quad (5)$$

The operator $A$ is self-adjoint, and its existence is guaranteed by the spectral theorem. Together with the dual weight $\tilde{\omega}$, it allows us to construct a semifinite trace on the crossed product[5].

**Lemma 1.** *Let $(M, G, \alpha)$ be a dynamical system satisfying the requirements in Theorem 2. The linear map $\tau : M \rtimes_\alpha G \to \mathbb{C}$ given by*

$$\tau(X) \equiv \tilde{\omega}(A^{-1}X) \quad (6)$$

*is a faithful semifinite normal tracial weight on $M \rtimes_\alpha G$.*

One can rewrite (6) in terms of the original weight $\omega$ by making use of the unnormalized "position eigenket" $|g\rangle$[6], as in

$$\tau(X) = \int_G \mu(g)\delta(g)^{-1}\omega\big(\langle e|X|g\rangle\big)\langle e|\lambda(\gamma(i))|g^{-1}\rangle, \quad (7)$$

where we denote formally $\lambda(\gamma(i))$ to be the self-adjoint generator of the left translation operator. We will refer to (7) as the *modular trace* on $M \rtimes_\alpha G$. We present a proof of Lemma 1 as well as a derivation of (7) in Appendix C.

We now give some examples where our main theorem applies and comment on the physical significance of these cases.

**Example 1** (Modular Group). *As our first example, consider $G = (\mathbb{R}, +)$, which acts on $M$ as the modular automorphism $\sigma$. $M \rtimes_\sigma \mathbb{R}$ is the familiar modular crossed product. $\mathbb{R}$ is unimodular with Haar measure $\mu(q) = dq$. The left translation operator is $\lambda(t) = e^{i\hat{p}t}$ where $\hat{p} \equiv -i\frac{d}{dq}$ is the momentum operator on $\mathbb{R}$. If $M$ is a factor, the modular crossed product admits a unique trace given by (7):*

$$\tau(X) = \int_\mathbb{R} dp\, e^{-p}\, \omega(\langle 0_q|X|p\rangle) = \omega\big(\langle 0_q|Xe^{-\hat{p}}|0_q\rangle\big),$$

*where we have written the $\mathbb{R}$ integral in momentum space[7]. This reproduces the formula given in [25, 28, 29].*

**Example 2** (Direct Product). *Consider $G = \mathbb{R} \times H$ where $\mathbb{R}$ is the modular automorphism group and $H$ is some locally compact group acting on $M$. This arises in physical systems such as in AdS-Schwarzchild where $H = SO(3)$ implements rotation in a fixed background [25]. Let $\mu(h)$ be the left Haar measure on $H$. The left translation operator is again $\lambda(t) = e^{i\hat{p}t}$ and since $G$ is a direct product, $\hat{p}$ commutes with $H$. This also allows us to decompose $|g\rangle \in G$ as $|g\rangle = |q, h\rangle$ with $q \in \mathbb{R}$ and $h \in H$. The trace can then be decomposed into a double integral, where the integral over $H$ trivializes and we are left with*

$$\tau(X) = \omega\big(\langle 0, e|Xe^{-\hat{p}}|0, e\rangle\big).$$

*The above trace reduces back to the previous case when $H$ is trivial.*

**Example 3** (Quantum Corner Symmetry). *Our theorem also holds for groups which are not direct products with $\mathbb{R}$. Let $G = SL_2(\mathbb{R}) \ltimes H_3$ where $H_3$ is the three-dimensional Heisenberg group. This group arises as a central extension of the Extended Corner Symmetry (ECS) group by $\mathbb{R}$ [78]. The ECS is the finite part of the Universal Corner Symmetry group [79]. These groups are relevant symmetries of quantum gravity with isolated corners. Working semiclassically, we may take $M$ to be the algebra of observables of matter and propagating gravitons in a subregion bounded by the corner. Then, supposing there is a weight $\omega$ that is invariant under $G$ and whose modular flow is implemented by the $\mathbb{R}$ central subgroup of $G$, the crossed product $M \rtimes_\alpha G$ is semifinite.*

---

[5] Unless $M \rtimes_\alpha G$ is a factor, there will exist multiple inequivalent traces. If it is a factor, there will exist a unique trace up to rescaling.

[6] The use of position eigenkets here may seem unorthodox, but can be made mathematically rigorous by the theory of rigged Hilbert spaces [76, 77].

[7] In the physics literature, the modular crossed product is often constructed in the *momentum basis* as opposed to the position basis used here. They are related by a simple swap $p \leftrightarrow q$.

## V. DISCUSSION

A natural generalization of Takesaki's result [24] on the semifiniteness of the modular crossed product of a Type III von Neumann algebra would be that the crossed product of such an algebra with *any* locally compact group containing the modular automorphism group is semifinite. In this letter, we have shown that such a statement is not necessarily true, revealing a deeper interplay between modular theory, crossed products, and physical algebras of observables in gauge theory and gravity. Our conclusion is that this is the case whenever $M$ admits a faithful, semifinite, and normal weight that is invariant under the group action and KMS with respect to the action of a subgroup. A surprising implication of this invariance is the centrality of the modular automorphism group in $G$. An almost converse statement of this result is obtained, namely that if we assume the modular automorphism of some weight is implemented in $G$, then the crossed product is semifinite *if and only if* the modular automorphism group is central provided the relative commutant of $M$ is trivial.

Two immediate questions arise from this analysis:

1. Is the centrality of the modular automorphism group in $G$ necessary for the semifiniteness of $N_\alpha := M \rtimes_\alpha G$ when $M$ is Type III? We aim to answer this question in future work by capitalizing on the following observation: take the crossed product of $N_\alpha$ itself with its modular automorphism group, $N_\alpha \rtimes_{\tilde\sigma} \mathbb{R}$. The resultant algebra tensor factorizes as $N_\alpha \otimes \mathcal{L}(\mathbb{R})$ if and only if $\tilde\sigma$ is inner, and thus $N_\alpha$ is semifinite.

2. How do we deal with the semifiniteness of crossed product algebras where the relevant symmetry group does not contain a $\mathbb{R}$ subgroup, or worse yet, has no center at all? Such a group arises in near horizon physics and JT gravity, where the symmetry is $SL_2(\mathbb{R})$ and has a discrete $\mathbb{Z}_2$ center [80, 81]. One possible resolution is to embed $G$ into a group $G'$ containing $\mathbb{R}$ as a central subgroup, however this may not be physically motivated. Another resolution is that maybe in those cases, the relevant physical algebra of observables is *not* the crossed product, but another subalgebra of the tensor product algebra $M \otimes B(L^2(G))$ invariant under a different extended automorphism.

In conclusion, our work has significantly extended the groups for which the crossed product $M \rtimes_\alpha G$ is semifinite when $M$ is type III. Moreover, our analysis has shown that the semifiniteness of crossed products with general locally compact groups is a subtle issue that is worth understanding to truly unravel the importance of crossed product algebras in physics.

### ACKNOWLEDGMENTS

We thank Ahmed Almheiri, Thomas Faulkner, Elliot Gesteau, Rob Leigh, and Antony Speranza for useful discussions.

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

## Appendix A: Haagerup form of the crossed product

Let $(M, G, \alpha)$ be a dynamical system, along with $\pi : M \to B(H)$ a faithful representation of $M$ on a Hilbert space $H$. We denote by $H_G = L^2(G, H)$ the canonical covariant representation space. The crossed product $M \rtimes_\alpha G$ can be obtained as a weak closure of Haagerup's algebra

$$M_G \equiv \{\mathfrak{X} : G \to M \mid \text{continuous, compactly supported}\}. \tag{A1}$$

Let $\rho : M_G \to B(H_G)$ be the representation induced by the canonical covariant representation $(H_G, \pi_\alpha, \lambda)$:

$$\rho(\mathfrak{X}) \equiv \int_G \mu(g) \, \lambda(g) \pi_\alpha(\mathfrak{X}(g)). \tag{A2}$$

Then, $M \rtimes_\alpha G = \rho(M_G)''$. In other words, $M_G$ is dense in the crossed product and thus we may establish results for $M \rtimes_\alpha G$ by first establishing it for $M_G$ and then carrying into the rest of the algebra by closure.

The set $M_G$ is made an involutive algebra by endowing it with a product $\star$ and an involution $\sharp$ which generalize the group algebra:

$$\begin{aligned}
(\mathfrak{X} \star \mathfrak{Y})(g) &\equiv \int_G \mu(h) \, \alpha_h\big(\mathfrak{X}(gh)\big)\mathfrak{Y}(h^{-1}), \\
\mathfrak{X}^\sharp(g) &\equiv \delta(g^{-1})\alpha_{g^{-1}}\big(\mathfrak{X}(g^{-1})\big)^*.
\end{aligned} \tag{A3}$$

Given (A3) we can define on $M_G$ an $M$-valued bilinear $\gamma : M_G \times M_G \to M$:

$$\gamma(\mathfrak{X}, \mathfrak{Y}) \equiv \big(\mathfrak{X}^\sharp \star \mathfrak{Y}\big)(e). \tag{A4}$$

Plugging (A3) back into (A4) one can show that

$$\big(\mathfrak{X}^\sharp \star \mathfrak{Y}\big)(e) = \int_G \mu(g) \, \mathfrak{X}(g)^* \mathfrak{Y}(g). \tag{A5}$$

Haagerup's operator valued weight $T : \rho(M_G) \to M$ is defined as

$$T\big(\rho(\mathfrak{X}^\sharp \star \mathfrak{X})\big) \equiv \gamma(\mathfrak{X}, \mathfrak{X}) = \int_G \mu(g) \, \mathfrak{X}(g)^* \mathfrak{X}(g). \tag{A6}$$

As mentioned, this defines an operator valued weight on all of $M \rtimes_\alpha G$ since $M_G$ is a dense subset. Eqn. (A6) can be interpreted as the $M$-valued generalization of the Plancherel weight. Given a faithful, semifinite and normal weight $\omega \in M_*$, the dual weight $\tilde\omega \in (M \rtimes_\alpha G)_*$ can be simply expressed in terms of $T$ as

$$\tilde\omega = \omega \circ T. \tag{A7}$$

## Appendix B: Modular automorphism of the Plancherel Weight

Consider the dynamical system $(\mathbb{C}, G, \beta)$ where $\beta$ is a trivial automorphism. Haagerup's algebra $\mathbb{C}_G$ is isomorphic to $C_c(G)$ – the set of continuous, compactly supported complex valued functions on $G$. This algebra acts on the Hilbert space $L^2(G)$, where the representation $\rho$ given in (A2) is the action by the group convolutional product. Thus, it is manifest that the crossed product in this case is isomorphic to the group von Neumann algebra: $\mathbb{C} \rtimes_\beta G = \rho(\mathbb{C}_G)'' \simeq \mathcal{L}(G)$.

A weight, $\phi$, on $\mathbb{C}$ is identified with a complex number

$$\phi(z) = \phi^* z, \ \phi \in \mathbb{C}. \tag{B1}$$

All such weights are tracial and thus have trivial modular automorphism, $\sigma_t^\phi(z) = z$ for all $z \in \mathbb{C}$. Eqn. (3) identifies dual weights simply as multiples of the Plancherel weight:

$$\tilde\phi\big(\rho(\mathfrak{X}^\sharp \star \mathfrak{X})\big) = \phi^* \gamma(\mathfrak{X}, \mathfrak{X}) = \phi^* \omega_G(\mathfrak{X}^\sharp \star \mathfrak{X}). \tag{B2}$$

Since the action $\beta$ is trivial the cocycles $u_t^{g,e} = \mathbb{1}$. Thus, we conclude from (4) that the modular operator associated with the Plancherel weight is simply $\left(\Delta_{\omega_G}^{it} f\right)(g) = \delta(g)^{it} f(g)$.

## Appendix C: A trace on the crossed product

In this appendix we work under the assumptions of Theorem 2. In particular, $(M, G, \alpha)$ is a dynamical system, $\gamma : \mathbb{R} \hookrightarrow G$ is an inclusion, and $\omega \in M_*$ is a faithful, semifinite, normal weight on $M$ which is quasi $\alpha$-invariant and KMS with respect to $\alpha \circ \gamma : \mathbb{R} \to \text{Aut}(M)$. The map

$$\tau : M \rtimes_\alpha G \to \mathbb{C}, \qquad \tau(X) \equiv \tilde\omega\big(A^{-1} X\big), \tag{C1}$$

will automatically be a faithful semifinite normal tracial weight on $M \rtimes_\alpha G$ provided $A \in M \rtimes_\alpha G$ implements the modular automorphism of the weight $\tilde\omega$ as $\tilde\sigma_t = \text{Ad}_{A^{it}}$. The fact that (C1) is a faithful, semifinite and normal follows immediately from the dual weight theorem and the invertiblility of $A$. The fact that it is tracial follows from the KMS condition

$$\begin{aligned}
\tau(XY) &= \tilde\omega\big(A^{-1} X Y\big) \\
&= \tilde\omega\big(A^{-1} X A A^{-1} Y\big) \\
&= \tilde\omega\big(\sigma_i(X) A^{-1} Y\big) \\
&= \tilde\omega\big(A^{-1} Y X\big) = \tau(YX).
\end{aligned} \tag{C2}$$

To construct the trace more explicitly, note that the dual weight can be written with Haagerup's operator valued weight (A7). Applying (A3) to the result we have

$$\begin{aligned}
\tau(X) &= \tilde\omega\left(A^{-1} X\right) \\
&= \omega\bigg(\big(\rho^{-1}(A^{-1}) \star \rho^{-1}(X)\big)(e)\bigg) \\
&= \int_G \mu(g)\, \omega\bigg(\alpha_g\big(\rho^{-1}(A^{-1})(g)\big) \cdot \big(\rho^{-1}(X)(g^{-1})\big)\bigg).
\end{aligned} \tag{C3}$$

We can explicitly write down the inverse map $\rho^{-1}$ by making use of the position eigenket $|g\rangle$[8] with a formal inner product $\langle h|g\rangle = \delta(h^{-1}g)$ and resolution of identity $\mathbb{1} = \int_G \mu(g)\,|g\rangle\,\langle g|$. For $\mathfrak{X} \in M_G$ we compute

$$
\begin{aligned}
\langle e \,|\, \rho(\mathfrak{X}) \,|\, g\rangle &= \int \mu(h)\,\langle e \,|\, \lambda(h)\pi_\alpha(\mathfrak{X}(h)) \,|\, g\rangle \\
&= \int \mu(h)\,\langle h^{-1}|g\rangle\,\alpha_{g^{-1}}(\mathfrak{X}(h)) \\
&= \alpha_{g^{-1}}(\mathfrak{X}(g^{-1})),
\end{aligned}
\tag{C4}
$$

where we have used the decomposition $\pi_\alpha(X) = \int \mu(g)\,\alpha_{g^{-1}}(X) \otimes |g\rangle\,\langle g|$. Thus, we obtain the relation $\mathfrak{X}(g) = \alpha_{g^{-1}}\left(\langle e \,|\, \rho(\mathfrak{X}) \,|\, g^{-1}\rangle\right)$ and we can write the trace as

$$
\begin{aligned}
\tau(X) &= \int \mu(g)\,\omega\left(\langle e|A^{-1}|g^{-1}\rangle\,\alpha_g(\langle e|X|g\rangle)\right) \\
&= \int \mu(g)\,\delta(g)^{-1}\omega\left(\langle e|X|g\rangle\right)\,\langle e|\lambda(\gamma(i))|g^{-1}\rangle.
\end{aligned}
\tag{C5}
$$

To move from the first to the second line we have used the quasi invariance condition of $\omega$ and the relation $A = \mathbb{1} \otimes \gamma(\lambda(i))$.

## Appendix D: Dual modular action on generators of the crossed product

In this appendix, we include the computation of the dual modular automorphism group of the dual weight $\tilde{\omega}$ on the generators of the crossed product $M \rtimes_\alpha G$.

1. The algebra representation:

$$
\begin{aligned}
[\tilde{\sigma}_t(\pi_\alpha(x))\xi](s) &= [\tilde{\Delta}^{it}\pi_\alpha(x)\tilde{\Delta}^{-it}\xi](s), \\
&= \alpha_{ts^{-1}}(x)\xi(s), \\
&= \pi_\alpha \circ \alpha_t(x)\xi(s), \\
&= \pi_\alpha \circ \sigma_t(x)\xi(s). \implies \tilde{\sigma}_t(\pi_\alpha(x)) = \pi_\alpha \circ \sigma_t(x).
\end{aligned}
\tag{D1}
$$

2. The group representation:

$$
\begin{aligned}
[\tilde{\sigma}_t(\lambda(g))\xi](h) &= [\tilde{\Delta}^{it}\lambda(g)\tilde{\Delta}^{-it}\xi](h), \\
&= \delta(h)^{it}\Delta_{h,e}^{it}[\lambda(g)\tilde{\Delta}^{-it}\xi](h), \\
&= \delta(g)^{it}\Delta_{h,e}^{it}[\tilde{\Delta}^{-it}\xi](g^{-1}h), \\
&= \delta(g)^{it}\Delta_{h,e}^{it}\Delta_{g^{-1}h,e}^{-it}\xi(g^{-1}h), \\
&= \delta(g)^{it}\pi\left(u_t^{h,g^{-1}h}\right)\xi(g^{-1}h), \\
&= \delta(g)^{it}\pi \circ \alpha_h^{-1}\left(u_t^{e,g^{-1}}\right)\xi(g^{-1}h), \\
&= [\delta(g)^{it}\pi_\alpha\left(u_t^{e,g^{-1}}\right)\lambda(g)\xi](h), \\
&= [\delta(g)^{it}\lambda(g)\lambda(g)^{-1}\pi_\alpha\left(u_t^{e,g^{-1}}\right)\lambda(g)\xi](h), \\
&= [\delta(g)^{it}\lambda(g)\pi_\alpha \circ \alpha_g^{-1}\left(u_t^{e,g^{-1}}\right)\xi](h), \\
&= [\delta(g)^{it}\lambda(g)\pi_\alpha\left(u_t^{g,e}\right)\xi](h), \implies \tilde{\sigma}_t(\lambda(g)) = \delta(g)^{it}\lambda(g)\pi_\alpha\left(u_t^{g,e}\right).
\end{aligned}
\tag{D2}
$$

We now use the above to prove a lemma useful in the proof of Theorem 2.

---

[8] Heuristically, $|g\rangle$ should be thought of having a "wave function" with support only at $g$.

**Lemma 2.** *Suppose the modular automorphism of a weight $\omega$, $\sigma^\omega$, commutes with the group action $\alpha : G \to Aut(M)$ on a von Neumann algebra, where $\alpha|_{\mathbb{R}} = \sigma$. Then, $\sigma_t = \alpha_{\gamma(t)}$ is inner implemented in $M \rtimes_\alpha G$.*

*Proof.* Beginning from the commutation theorem, we have

$$\alpha_g \otimes \mathrm{Ad}_{\lambda_r(g)}(X) = X \tag{D3}$$

for any element $X \in M \rtimes_\alpha G$. Acting with $\alpha_g^{-1} \otimes I$ from the left gives

$$I \otimes \mathrm{Ad}_{\lambda_r(g)} = \alpha_{g^{-1}} \otimes I. \tag{D4}$$

Consider the action of $\sigma_t = \alpha_{\gamma(t)}$ on the two generators of the crossed product

$$(\sigma_t \otimes I)\left[\lambda(g)\right] = (I \otimes \mathrm{Ad}_{\lambda_r(\gamma(-t))})(\lambda(g)) = \lambda(g), \tag{D5}$$

where we have used that the left and right regular representations commute, and

$$\begin{aligned}
\left[(\sigma_t \otimes I)\pi_\alpha(x)\xi\right](g) &= (I \otimes \mathrm{Ad}_{\lambda_r(\gamma(-t))})(\pi_\alpha(x)\xi)(g), \\
&= (I \otimes \mathrm{Ad}_{\lambda(\gamma(t))})(\pi_\alpha(x)\xi)(g), \\
&= (I \otimes \mathrm{Ad}_{\lambda(\gamma(t))})(\pi_\alpha(x)\xi)(g), \\
&= (\pi_\alpha \circ \sigma_t(x)\xi)(g),
\end{aligned}$$

where we have used that $\lambda_r(\gamma(t)) = \lambda(\gamma(-t))$ under the assumption that $\mathbb{R} < G$ is central. This may be verified easily on an element $\xi \in L^2(G, H)$. Note that we can write the above two results as

$$(\sigma_t \otimes I)\left[\lambda(g)\right] = \lambda(\gamma(t)g\gamma(-t)), \quad (\sigma_t \otimes I)\pi_\alpha(x) = \lambda(\gamma(t))\pi_\alpha(x)\lambda(\gamma(-t)). \tag{D6}$$

We see that $\mathrm{Ad}_{\lambda(\gamma(t))}$ and $\sigma$ agree on the two generators of the crossed product $M \rtimes_\alpha G$ and so are identified on the entire algebra. The former is manifestly inner, and so we are done. $\qquad\square$

### Appendix E: Inner implementer of the dual automorphisms

In this appendix, we show that when $\pi_\alpha(M)^c = \mathbb{C}I$ and $\omega$ is KMS with respect to $\alpha \circ \gamma : \mathbb{R} \to Aut(M)$, then $\lambda(\gamma(t))$ is the unique inner implementer of the dual automorphism of $\omega$. Indeed, suppose that there exists some a family of unitary operators $Q_t \in M \rtimes_\alpha G$ such that

$$\tilde\sigma = \mathrm{Ad}_{Q_t}. \tag{E1}$$

Then, we may constrain the properties of $Q_t$ by looking at the product of two generators

$$\begin{aligned}
\tilde\sigma(\pi_\alpha(x)\lambda(g)) &= Q_t\pi_\alpha(x)\lambda(g)Q_{-t}, \\
&= Q_t\pi_\alpha(x)Q_{-t}Q_t\lambda(g)Q_{-t}, \\
&= \lambda(\gamma(t))\pi_\alpha(x)\lambda(\gamma(t))^{-1}Q_t\lambda(g)Q_{-t}.
\end{aligned} \tag{E2}$$

This is equivalent to

$$Q_{-t}\lambda(\gamma(t))\pi_\alpha(x)\lambda(\gamma(t))^{-1}Q_t = \pi_\alpha(x) \implies X_t := Q_{-t}\lambda(\gamma(t)) \in \pi_\alpha(M)' \cap M \rtimes_\alpha G. \tag{E3}$$

The newly defined family of unitary operators $X_t$ must then lie in the relative commutant of $\pi_\alpha(M)$ for $Q_t$ to innerly generate the modular flow. Under the assumption, $X_t$ must be proportional to the identity implying that $Q_t = \lambda(\gamma(t))$ up to a scalar. In other words, in this case, $\mathrm{Ad}_\lambda$ is the unique (up to scaling) inner implementer of the dual modular automorphism.