# Peer review of "Semifinite von Neumann algebras in gauge theory and gravity"

_SciPost Physics_

## Round 1 · Referee Report · Anonymous · 2024-8-9

Report

This is essentially a math paper, but with important applications to physics in mind. The "crossed product" algebraic structure has proven important in understanding the implementation of constraints in gauge theory at the level of operator algebras. In certain limited settings within gravity, a theorem says that the crossed product algebra is "semifinite," meaning renormalized entropy differences can be defined. As the first step in an effort to expand this prior work, the present paper purports to uncover a general condition under which a crossed product algebra is semifinite. This is an important problem, and I think the paper is probably correct. However, I say "probably" because many details are missing and I cannot fully judge the quality of the argument. The paper is extremely technical, and does not include enough exposition on the technical tools for the paper to be assessed on its own merit without a lengthy dive through related mathematical literature.

Just as a taste of what I mean, in the first few sentences of section III, the phrase "faithful, semifinite, normal weight" is introduced with no explanation. This is an important concept in the mathematical literature, but has not sufficiently penetrated the physics literature to be used without explanation. They also immediately start to talk about the relative modular operator without explaining what this is --- in this case the complaint is more than semantic, because non-state weights do not have relative modular operators, even though they do have relative modular flow. A few sentences later the authors comment on "a generalization of the group algebra" without saying what "the group algebra" is, and start writing the modular function \delta associated with a Haar measure without explaining what it is. On the next page they say "the construction of the dual weight is originally due to Digernes for the separable case," but do not say what "the separable case" is or what it means.

I am finding it very difficult to give this paper a complete review without dedicating an inordinate amount of time to unearthing all of the technical details myself. I request that the authors resubmit a version of the paper with additional exposition and context for the technical work that they have accomplished, as the work seems important and worth understanding, but is not suitable for review or publication in a physics journal in its current form.

I also request that while the authors are undertaking this revision, they make sure that all of the paper's important claims are clearly stated in the introduction. For example, they discuss a condition under which an algebra is semifinite, but it is not clear to me whether they have conditions under which the associated algebra is a factor, which is an important ingredient in existing work.

Recommendation

Ask for major revision

  • validity: -
  • significance: -
  • originality: -
  • clarity: -
  • formatting: -
  • grammar: -

Author:  Shadi Ali Ahmad  on 2024-09-15  [id 4781]

(in reply to Report 1 on 2024-08-09)

We thank the referee for the time and effort put into preparing this report. While we concede that our paper may be regarded as more on the technical side of the physics literature, we note that we have submitted our work under both High Energy Theory and Mathematical Physics for precisely the reason the referee identified. Our mathematical result is relevant for physical applications. We also thank the referee for the kind words about the importance of our work.
 We have revised parts of our work and submitted a new version according to the referee’s comments, namely:
 (1) Included a new section titled “Preliminaries on modular theory” which contains a review of the relevant aspects of von Neumann algebras, their weights and states, and relative modular theory required to understand the main result of our paper. We refer the interested reader to four works from different perspectives if they would like to learn even more.
 (2) Defined the module function of a group where it first appears on page 4.
 (3) Clarified what is meant by the separable case in Digernes' original work on crossed product algebras in footnote 7.
 (4) Added exposition in footnote 8 on the status of the question the referee posed in the final paragraph regarding the conditions for which the crossed product is a factor.
 (5) Provided a sentence in Appendix A on the definition of the group algebra, and removed the phrase “generalization of the group algebra” in the main text as it is strictly not necessary to understand what the group algebra is either to understand our work or even motivate the definition of Haagerup’s algebra.
 We hope that the changes made above have made our manuscript a more self-contained unit and improved its readability and presentation. We thank the referee again for their comments.

---

## Round 1 · Referee Report · Anonymous · 2024-9-12

Report

\begin{center}
{Report on Semifinite von Neumann algebras in gauge theory and gravity}
\end{center}
First of all let me apologize for the delay due in part to the summer Holidays.

This is an interesting and potentially deep paper. If I understand correctly the key result of the paper is the extension of the classic Takesaki results on duality and crossed products to general groups $G$ locally compact but {\it not abelian}. Most concretely, in the author own words: "to provide a sufficient condition for the semifinitiness of the crossed product $M\rtimes_{\alpha} G$ for $G$ locally compact but general". Is this interpretation correct ?

In a nutshell let us present the paper in the following terms. Let us assume $M$ is a type $III_1$ factor but {\it injective}. After the seminal work of Connes and Haagerup we know that $M$ is unique and isomorphic to Araki-Wood $R_{\infty}$ factor. Moreover $M$ can be represented as $N\rtimes_{\theta}R$ with $N$ semifinite with trace $\tau$ satisfying $\tau\theta_s=e^{-s}\tau$. Moreover for $M$ injective is isomorphic to $P\otimes F_{\infty}$ for $F_{\infty}$ a type $I_{\infty}$ factor and $P$ isomorphic to $M$. For a dominant weight $\omega$ on $M$ we have $\omega=\phi\otimes Tr$ and the representation of the centralizer $M_{\omega}$ as $P\rtimes _{\sigma_{\phi}}R$ which is a type $II_{\infty}$ factor. The authors want to generalize this result on $M_{\omega}$ using qusi invariant weights in the sense of theorem 2.

This is an interesting attempt but I will need before accepting the paper to ask the authors the following question:

Q: Are the weights satisfying the conditions of Theorem 2 of the paper integrable weights?

My problem is that if the answer is yes then the results of this paper are just the ones already in Takesaki original paper.

Could the authors elaborate a bit on the former question Q? Note that this question in intimately related with the fact that injective type $III_1$ are unique.

Recommendation

Ask for minor revision

  • validity: good
  • significance: good
  • originality: good
  • clarity: high
  • formatting: good
  • grammar: perfect

Author:  Shadi Ali Ahmad  on 2024-09-15  [id 4782]

(in reply to Report 2 on 2024-09-12)

We thank the referee for their generous comments and question. Indeed, their interpretation of our main result is correct. What we accomplished in this paper is the generalization of Takesaki's seminal result identifying the crossed product of a type III$_{1}$ factor with its modular automorphism group as a semifinite von Neumann factor. Our unique contribution is to identify the conditions under which this statement continues to hold when the group in question is no longer the modular automorphism group, but rather any general locally compact group. For clarity, to reproduce Takesaki's result using our work, one could simply set $G = R$, the additive group of real numbers, in our Theorem 2 and see that the conditions automatically hold. Namely: (1) $R$ is central in itself (since it is Abelian) and (2) the defining KMS weight is invariant under its modular flow. However, our result applies to more general G than just this choice, as shown in later examples.

The presentation of our paper by the referee seems to only refer to the crossed product of $M$ with $R$, which is not the case we have primarily concentrated on. While it is true that a Type III$_{1}$ factor $M$ can be written as a crossed product of the factor $N = M \rtimes_\sigma R$ with $R$, we rather ask the question of whether or not $M \rtimes_{\alpha}G$ itself is semifinite. In the event that this algebra is not semifinite, one can still take the crossed product of $M \rtimes_{\alpha}G$ with its modular automorphism group, which would be appealing to Takesaki’s result, to obtain a semifinite factor. In our work, we give sufficient conditions for when $M \rtimes_{\alpha} G$ is semifinite on its own merit without the need to appeal to this further step.

Addressing the reviewer's question we interpret `integrable weights' in this context to refer to either (1) amenability or (2) finiteness of weights under taking the Haar average. If this is an appropriate interpretation, then the answer to question Q is no for our paper (and for Takesaki’s original result, for that matter) since the groups we consider are a priori locally compact but not compact. We would moreover like to emphasize that at no point in the text have we relied upon assumptions about the injectivity of the factors involved.

We believe that these considerations should resolve the reviewer's concerns that our result is merely a restatement of Takesaki's. This is further evidenced by our examples. We would also like to emphasize that our general result, besides extending the mathematical content of Takesaki's work, also has an elevated physical context. Being able to track the semifiniteness of crossed product algebras involving generic locally compact groups allows for the diagnosis of entanglement divergences in theories with arbitrary grouplike symmetries. This is in contrast with Takesaki's result which is limited to the modular automorphism and therefore cannot be applied to general symmetry groups.

We thank the referee again for their comments.

---

## Editorial Decision

resubmitted